# Development and Characterization of a Subcutaneous Implant-Related Infection Model in Mice to Test Novel Antimicrobial Treatment Strategies

**DOI:** 10.3390/biomedicines11010040

**Published:** 2022-12-24

**Authors:** Charlotte Wittmann, Niels Vanvelk, Anton E. Fürst, T. Fintan Moriarty, Stephan Zeiter

**Affiliations:** 1AO Research Institute Davos, 7270 Davos, Switzerland; 2Equine Department–Vetsuisse Faculty, University of Zurich, 8057 Zurich, Switzerland

**Keywords:** implant-related infection, mouse model, biomaterial testing

## Abstract

Orthopedic-device-related infection is one of the most severe complications in orthopedic surgery. To reduce the associated morbidity and healthcare costs, new prevention and treatment modalities are continuously under development. Preclinical in vivo models serve as a control point prior to clinical implementation. This study presents a mouse model of subcutaneously implanted titanium discs, infected with *Staphylococcus aureus*, to fill a gap in the early-stage testing of antimicrobial biomaterials. Firstly, three different inocula were administered either pre-adhered to the implant or pipetted on top of it following implantation to test their ability to reliably create an infection. Secondly, the efficacy of low-dose (25 mg/kg) and high-dose (250 mg/kg) cefazolin administered systemically in infection prevention was assessed. Lastly, titanium implants were replaced by antibiotic-loaded bone cement (ALBC) discs to investigate the efficacy of local antibiotics in infection prevention. The efficacy in infection prevention of the low-dose perioperative antibiotic prophylaxis (PAP) depended on both the inoculum and inoculation method. Bacterial counts were significantly lower in animals receiving the high dose of PAP. ALBC discs with or without the additional PAP proved highly effective in infection prevention and provide a suitable positive control to test other prevention strategies.

## 1. Introduction

Orthopedic-device-related infection (ODRI) represents one of the most severe complications in orthopedic surgery. Infection results in increased postoperative pain, delayed healing, loss of function and the potential amputation of the affected limb [1]. Furthermore, it represents a significant socioeconomic burden causing a substantial increase in cost and healthcare resource utilization when compared to non-infected cases [2,3]. Infection rates vary between 0.5 and 40% depending on various factors including host physiology (e.g., diabetes mellitus) and injury characteristics (e.g., Gustilo–Anderson type) [4,5,6]. Among isolated bacteria from ODRI, *Staphylococcus aureus* (*S. aureus*) are most prevalent and almost exclusively used in preclinical studies of ODRI [7,8]. The presence of foreign material, such as a fracture fixation device, has been described as an independent risk factor for infection [4]. Elek et al. showed that the presence of sutures in human volunteers decreased the minimal infecting dose with a *Staphylococcus* strain from 5 × 10^6^ to 3 × 10^2^ [9]. In a guinea pig model, Zimmerli et al. found the minimal infecting dose of *Staphylococcus aureus* (*S. aureus*) in the presence of tissue cages to be 1 × 10^2^ colony-forming Units (CFU), while in the absence of any foreign material 1 × 10^8^ CFU did not reliably cause an infection [10]. These studies thus demonstrate the important contribution of a foreign body in infection development.

In orthopedic surgery, the use of perioperative antibiotic prophylaxis (PAP) is a common practice to decrease the incidence of postoperative infection [11]. However, impaired vascularization resulting from the initial trauma and biofilm formation on the foreign body could prevent a sufficiently high antimicrobial concentration at the surgical site. Increasing antibiotic doses raises the risk of systemic toxicity and side effects. Local application of antibiotics directly to the surgical site can support prophylaxis and can reach a higher concentration with minimal systemic side effects [12]. Polymethylmethacrylate (PMMA) and collagen sponges are commercially available carriers for local antibiotic delivery and are currently used routinely in orthopedic surgery to prevent ODRI [13]. Disadvantages of these devices are the exothermic process of PMMA polymerization which poses the risk of thermal necrosis of the bone and prohibits the incorporation of heat sensitive antibiotics [14]. Additionally, PMMA is not biodegradable and has to be removed at a later surgical revision.

Preclinical in vivo models serve as a critical control point prior to the translation of any new procedure or intervention. These models can be grouped according to their complexity [15], with one option being to make animal models as clinically relevant and translational as possible, whereby the clinical scenario ought to be replicated as closely as possible [8]. Fully mimicking clinical conditions, however, requires a high level of expertise, appropriate equipment, and may also increase burden upon the animal and may not be required for all studies. For these reasons, models of high complexity are not justifiable for every stage of medical device development and less complex animal models can be utilized for early screening or proof of concept studies. For example, placing an implant subcutaneously in a non-functional position is a simple procedure and renders the implant as a foreign body, without any effort to function as an orthopedic device. Placing an implant subcutaneously does, however, include the host’s response to novel biomaterials that may be sufficient at an early stage of product development or proof of concept studies. 

Previous research by our own group, evaluating an antimicrobial coating by placing the coated implant in a subcutaneous pocket on the back of mice, revealed the lack of a fully characterized and standardized murine model. The impact of a single injection of antibiotics prior to surgery was not evaluated in that study (unpublished data).

This study presents the design and characterization of a subcutaneous implant-associated infection model in mice to fill that gap in the early-stage testing of antimicrobial biomaterials. The primary objective was to determine the optimal bacterial dose and inoculation method to establish a reliable *S. aureus* infection. The secondary objective was to investigate the efficacy of systemic and local antibiotics in infection prevention using clinical standard prophylactic antibiotics and commercially available antibiotic releasing biomaterials. 

## 2. Materials and Methods

### 2.1. Study Overview

The study was approved by the Ethical Committee of the Canton of Grisons, Switzerland (TVB 20_2019 and 02E_2020). All procedures were performed in an Association of Assessment and Accreditation of Laboratory Animals Care International (AAALAC)- approved facility and according to the Swiss animal protection law and regulation.

Sixty-five female C57BL/6N mice (Charles River Laboratories, Germany) were included across three study phases (Table 1 summarizes groups and number of animals per group). 

The first phase (dose-finding phase, group 1) aimed to determine the optimal route of administration and bacterial inoculum required to reliably create an infection. One titanium implant was inserted subcutaneously on either side of the spine. Bacteria were introduced either pre-adhered to the implant surface or an equal number were pipetted in suspension on top of the implant once placed in the tissue pocket. Bacteria were given in one of three bacterial doses (High, 1 × 10^6^ CFU/mL; Medium, 1 × 10^4^ CFU/mL; Low, 1 × 10^3^ CFU/mL) (Figure 1). The surgeon was blinded to the route of administration until pipetting was performed or the pre-inoculated implant was presented. Afterwards, the animals were observed for three days before scheduled euthanasia.

In the second phase (systemic antibiotic prophylaxis phase, group 2 and 3), the efficacy of two different doses of systemically administered cefazolin (Labatec-Pharma SA) in infection prevention was evaluated to mimic the clinical situation where PAP is routinely started prior to surgery. The first dose (25 mg/kg) was extrapolated from clinical practice, where administration of 2 to 3 g (25 mg/kg to 37.5 mg/kg) is recommended. The second dose (250 mg/kg) was based on research from Stavrakis et al., showing increased antimicrobial efficacy in an implant-associated infection model in mice with high dose antibiotics (200 mg/kg) [16]. A single injection of antibiotics was injected subcutaneously approximately 15 min prior to surgical incision. Afterwards, animals were infected as in the dose-finding phase (either pre-adhered to the implant surface or an equal number were pipetted).

In the local antibiotic prophylaxis phase (groups 4 to 6), antibiotic-loaded bone cement (ALBC) discs were implanted instead of titanium implants unilaterally on either the left or right side of the spine to study the efficacy of local antibiotics in infection prevention. The side of the implant was randomly assigned. Immediately after implantation, the lowest inoculum to reliably cause an infection in the dose-finding phase was pipetted onto the implant. A proportion of the animals received additional systemic antibiotics prior to implantation of the ALBC disc. Four additional animals received one titanium implant with the same inoculum and functioned as a control group.

The surgeon as well as the person collecting and analyzing data (CFU and body weight) were blinded. Animals were operated in phases based on the study design mentioned above on seven different surgery days (6 to 12 animals/day). Animals within the same cage were randomly allocated to the predefined systematically rotated group assignment.

### 2.2. Animal Welfare and Ethical Approvals

Female C57BL/6N mice (Charles River Laboratories, Germany) of 13–19 weeks of age were included in the study. The mean weight at inclusion was 233 ± 1.3 g. The animals were housed in groups of three to five animals in individually ventilated cages (Techniplast and Allentown, 530 cm^2^ ground floor) with a 12 h light/dark cycle (7 a.m. to 7 p.m.). They were allowed ad libitum access to autoclaved tap water and food (Kliba Nafag, Provimini Kliba SA, 3436 EXS12 S/R Entretien Extrude, Alleinfuttermittel für Mäuse und Ratten). Postoperative monitoring consisted of scoring certain parameters twice a day by a veterinarian. These included behavior, external appearance, breathing, excretions, wound healing, and bodyweight (Appendix A).

### 2.3. Implants

The metal implants used in this study were composed of medical grade titanium alloy niobium (TAN Grade NB, L. Klein SA, Switzerland), and were 2 mm thick with a diameter of 6 mm. Further processing involved cleaning, degreasing and rotofinishing with a particle size of 3 × 2.5 × 3 mm. Finally, the surface was anodized (TioCol™ KKS Ultraschall AG, Frauholzring 29, 6422 Steinen, Switzerland) to a gold-colored finish.

ALBC was prepared to the same dimensions as the titanium discus using a custom-made mold (Figure 1). The bone cement was made of PMMA (Palacos^®^ R+G, Heraeus Medical, Philipp-Reis-Straße 8-13, 61273 Wehrheim, Germany), loaded with gentamicin (12.25 mg/g bone cement) and prepared according to the manufacturer’s guidelines.

### 2.4. Surgical Intervention and Euthanasia

The surgical procedure was performed under general anesthesia and analgesia. Induction of anesthesia was achieved by exposing the animals to 8% sevoflurane in 100% oxygen after transferring them to an induction box. Sevoflurane was reduced to approximately 2.5% in 100% oxygen to maintain anesthesia. Oxygen flow was set to 700 mL/minute during the surgical procedure. Preoperative analgesia included subcutaneous injections of buprenorphine (0.03 mg/kg, Bupaq^®^ ad us. vet., Injektionslösung; Streuli Pharma AG, Bahnhofstrasse 7, 8730 Uznach, Switzerland) and carprofen (5 mg/kg, Rimadyl^®^ ad us. vet., Injektionslösung; Zoetis Schweiz GmbH, Rue de la Jeunesse 2, 2800 Delémont, Switzerland). One ml of prewarmed Lactated Ringer‘s solution (Ringer Spüllösung Ecobag^®^ 250 mL; B. Braun Medical AG, Seesatz 17, 6204 Sempach, Switzerland) was injected subcutaneously between the shoulder blades for temperature management and to replace perioperative fluid loss. The surgical field was clipped from the tail-base to the neck and aseptically prepared with Hibiscrub^®^ (chlorhexidindigluconat 40 mg per 1 mL; CPS Cito Pharma Service, Gschwaderstrasse 35/D, 8610 Uster, Switzerland) and 96% ethanolum (Softasept N; B. Braun Medical AG, Seesatz 17, 6204 Sempach, Switzerland). The animal was then positioned on the surgery table and draped with Dermadrape^®^ (Tiaset). On each side, an incision of 0.4 cm was made 1 cm lateral to the spine, avoiding bridging between sites. Subsequently, the implants were positioned subcutaneously with minimal tissue damage, taking care not to contaminate the implant during placement. Depending on the group, bacteria were pre-adhered onto the implant or pipetted onto their surface directly after implantation. The subcutis was closed with simple interrupted sutures using 6-0 Monocryl (Monocryl^®^, Ethicon, Johnson & Johnson AG, Gubelstrasse 34, 6300 Zug, Switzerland). The skin was closed with an intradermal suture using 5-0 Vicryl (Vicryl^®^, Ethicon, Johnson & Johnson AG, Gubelstrasse 34, 6300 Zug, Switzerland). Postoperative analgesia included paracetamol (acetaminophen) of 1.9 mg/mL added to the drinking water for 3 days. Body temperature was maintained during preparation and surgery using heating pads, and all animals received eye ointment. 

### 2.5. Bacteria and Inoculum Preparation

A clinical *S. aureus* strain (JAR060131), isolated from a patient with an infected hip prosthesis, was used [17]. The strain is broadly antibiotic susceptible, except for resistance to penicillin. It is available at the Swiss Culture Collection, with accession number CCOS 890. The bacterial inocula were individually prepared in phosphate-buffered saline solution (PBS, Merck KGaA, Frankfurterstrasse 250, 64293 Darmstadt, Germany) for each surgery. Three different bacterial inocula were used in this study: high, 1 × 10^6^ CFU/mL; medium, 1 × 10^4^ CFU/mL; and low, 1 × 10^3^ CFU/mL. A quantitative culture of each inoculum was performed immediately after preparation to check the accuracy of the prepared inoculum. The inoculum was pre-adhered by dipping the implant in the inoculum or 20 µL of inoculum was pipetted on top of it after implantation.

### 2.6. Quantitative Bacteriology at Euthanasia

All animals were kept at 4 °C after euthanasia and dissected within 4 h. At dissection, the surgical wound was first inspected followed by dissection of the implant and surrounding soft tissue. Post-mortem quantitative bacterial cultures were performed in all animals for the soft tissue adjacent to the implant and the implants itself. Soft tissue samples were homogenized in PBS using a homogenizer (Omni TH, tissue homogenizer TH-02/TH21649) until the tissue was a fine suspension and implants were sonicated in a Bandelin Ultrasonic water bath (Model RK 510 H) for 3 min. Serial tenfold dilutions of both solutions were plated on Tryptic soy agar (TSA) plates (Liofilchem srl, Via Scozia, 64026 Roseto degli Abruzzi TE, Italy). Bacterial growth was checked to determine if it was *S. aureus* using the latex agglutination test (Staphaurex™, Thermo Fisher Scientific Inc, Neuhofstrasse 11, 4153 Reinach TechCenter, 4153 Basel, Switzerland). The TSA plates were kept at room temperature for an additional 24 h to check for any slow-growing contaminants.

### 2.7. Statistical Analysis

Statistical analysis was performed using the Kruskal–Wallis test for comparisons between multiple groups. In the case of a statistically significant difference, multiple comparisons were performed using Dunn’s multiple comparisons test. The Mann–Whitney test was used for comparing two groups. 

## 3. Results

### 3.1. Animal Welfare

All animals recovered uneventfully from general anesthesia (average duration: 41 min) following surgery (average duration: 21 min). For the first 24 h, the animals had a score of up to three which declined thereafter. No difference between groups was observed. One animal of the dose-finding phase had to be excluded due to wound dehiscence with the subsequent exposure of the implant, but otherwise, all animals reached the scheduled euthanasia timepoint and showed no clinical signs of systemic infection. 

In the dose-finding phase, animals infected with the low bacterial inoculum gained 0.9 ± 1.2 g, while animals infected with the medium and high inoculum lost 0.8 ± 1.1 g and 0.3 ± 0.9 g, respectively. 

Animals included in the systemic antibiotic prophylaxis phase and treated with a low dose of systemic antibiotics lost 0.5 ±1.0 g when infected with the low inoculum and 0.3 ±0.9 g when infected with the medium inoculum. In the group receiving the high dose of systemic antibiotics, mice lost 1.3 ± 2.3 g. Animals being treated only with ALBC lost 1.2 ± 0.4 g, while the animals treated with a combination of local and systemic antibiotics lost 0.8 ± 0.7 g. Mice in the control group lost 1.2 ± 0.2 g. There was no significant difference in weight change between the groups within the different study phases.

### 3.2. Bacteriology

#### 3.2.1. Dose Finding

The goal of this phase was to determine the optimal route of administration and bacterial inoculum required to reliably create an infection. None of the 19 mice included in the dose-finding phase were able to eradicate the infection (Figure 2). For both inoculation methods, CFU counts of soft tissue samples were significantly higher in animals receiving the high bacterial inoculum when compared to animals receiving the low inoculum (*p* = 0.021 for pipette inoculation, *p* = 0.004 for disc inoculation). A similar trend was observed in the CFU counts recovered from the surface of the implants, but this was only significant in the disc inoculation method (*p* = 0.050 for medium versus low inoculum).

CFU counts were generally comparable between the different inoculation methods. However, for the medium inoculum CFU, counts in soft-tissue samples were higher after disc inoculation (*p* = 0.0368). In this first phase, all three inocula and two inoculation methods reliably created an infection, with lower bacterial counts in the low inoculum groups.

#### 3.2.2. Systemic Antibiotic Prophylaxis

This phase aimed to investigate the efficacy of systemic antibiotics in the prevention of infection. CFU counts of both the soft tissue and implant samples were comparable between the animals receiving the low and medium inoculum and being treated with 25 mg/kg of cefazolin. A significant difference in CFU counts between animals infected with the low inoculum and treated with either 25 mg/kg or 250 mg/kg of cefazolin was found in both sample types after disc inoculation (*p* = 0.020 for implant sample, *p* = 0.010 for soft tissue sample). For this phase of the trial, no significant difference in CFU counts was found between pipette inoculation and disc inoculation. In one out of eight animals receiving 250 mg/kg of cefazolin subcutaneously prior to surgery, no bacterial growth was detectable. The other mice were not able to completely clear the infection. Both the implant and soft tissue samples remained negative in one of the mice after pipette inoculation, while another mouse was able to clear the infection of both samples after disc inoculation (Figure 3).

#### 3.2.3. Local Antibiotic Prophylaxis

Control animals receiving the titanium implant without any systemic antibiotics presented with a high CFU count on all samples. In the group receiving no systemic antibiotic prophylaxis, bacterial growth was observed in three soft tissue samples. ALBC discs were not infected. In the group of mice receiving additional systemic antibiotic prophylaxis, one animal was not able to eradicate the infection in both the soft tissue and implant samples. All other animals cleared the infection. There was no statistically significant difference in CFU count between the groups with ALBC alone and ALBC combined with subcutaneous cefazolin (Figure 4).

## 4. Discussion

This study presents the characterization of a low complexity subcutaneous implant-associated infection model in mice. This model includes the host’s response to novel biomaterials and can be applied in future research on future strategies in the prevention and treatment of ODRI.

The dose-finding phase of this study revealed that an *S. aureus* bacterial inoculum of 1 × 10^3^ CFU/mL was sufficient to create a reproducible infection. In in vivo testing of antimicrobial coatings, it is critical that an appropriate bacterial burden is introduced to the surgical site. Bacterial inocula used in dose-finding studies typically vary between 1 × 10^2^ and 1 × 10^8^ CFU/site [18]. If the bacterial burden is too high, it does not adequately reflect the clinical situation of sterile surgery, where bacterial numbers are in the low hundreds and mostly airborne [19,20]. Furthermore, high inocula may mask the ability of a novel coating to prevent infection due to the overwhelming bacterial numbers typically not present in any clinical scenario. In contrast, the 1 × 10^3^ CFU/mL dose established in the present study produced a sufficient but variable infection in all animals without leading to sepsis, making this a safe and suitable bacterial dose for future biomaterial testing.

This study also investigated different methods of bacterial administration, representing the different routes through which surgical implants might become infected during surgery. Pre-adhered bacteria reflect implants having been contaminated on the instrument tray by means of circulating airborne bacteria [20], whereas the pipetted bacterial inoculum represents bacteria being transferred during surgery or entering the wound from the skin soon afterwards. Generally, both methods produced comparable CFU counts, and a difference was only found with the medium inoculum in the dose-finding phase, where inoculation with pre-adhered bacteria resulted in higher CFU counts. The validity of both methods allows the researcher to adapt the protocol according to their needs. Pipetting the inoculum provides the advantages of negating implant preparation and the ability to test coated materials; however, containing the bacterial suspension within the surgical site during closure can be challenging. Pre-inoculated implants provide the unique advantage of reducing surgical time and staff but requires special care while handling to not disturb the bacterial layer and it cannot be performed on implants with antimicrobial coatings. Both approaches create a reliable infection for use in a wide range of research applications. 

In the second phase of this study, the efficacy of PAP in infection prevention was investigated in this model to mimic the common practice of PAP in orthopedic surgery to decrease the incidence of postoperative infection. In a relevant model of ODRI, PAP alone would decrease the bacterial burden, but not avoid/clear the infection in all cases. In human medicine, the recommended dose for prophylactic cefazolin administration varies from 2 to 3 g (25 mg/kg to 37.5 mg/kg in obese patients) [11]. In this model, monotherapy with a 25 mg/kg dose of cefazolin was not effective in clearing the infection; however, significantly lower CFU counts were found after this treatment for infections created with the medium inoculum. In this way, the model reflects the clinical situation where infection often develops despite systemic antibiotic prophylaxis. Since the dose extrapolated from clinical practice was not effective in eliminating bacterial burden, the antimicrobial dose was increased to 250 mg/kg. However, this was still ineffective in completely clearing the infection and a significant difference in CFU counts when compared to treatment with the 25 mg/kg dose of cefazolin after infection development with the low inoculum was found only after disc inoculation. Species differences could account for the discrepancy between the required antimicrobial dose. Firstly, the serum half-life of cefazolin in mice is much lower than in humans (~2 h in humans vs. ~23 min in mice) [11,21]. Secondly, serum protein binding is higher in humans, leaving mice with a higher component of unbound and active drug [16]. These factors could explain the limited effect on infection prevention of systemic antibiotics monotherapy in this study. This shows that for future research in mice intravenous administration of systemic antibiotics might be preferable over subcutaneous administration. 

For the successful translation of new interventions, a clinically relevant preclinical model needs to include a positive control reflecting the current state of the art to judge the effectiveness of the new treatment modality in comparison. Therefore, in the third phase of this preclinical study, locally applied gentamicin impregnated PMMA was included, as it is currently used routinely in orthopedic surgery to prevent ODRI. In soft tissue samples, the gentamicin-loaded bone cement completely cleared the infection in seven animals and markedly reduced the bacterial burden of the remaining three animals. The implants of all ten animals were culture negative. All samples were tested for gentamicin resistance and proven negative. Therefore, the local application of gentamicin using a PMMA carrier provides a significant advantage over the sole use of systemic PAP and provides a benchmark for the future testing of new materials. 

Following previously conducted research, where an additional systemic antimicrobial injection in combination with gentamicin-loaded bone cement resulted in the lowest revision rate of total hip arthroplasty [22], an additional cefazolin injection (250 mg/kg) was given prior to surgery. The combination of systemic antibiotic prophylaxis as well as ALBC implants was highly efficacious in reducing the bacterial burden. However, one animal was highly infected in both obtained samples. No bacteria could be cultured on the implants of the remaining animals. Samples were checked for resistance against gentamicin with negative results. A possible explanation for this is an ineffective eradication of bacteria with a subsequent recolonization or contamination of the implant at the time of harvesting. Nonetheless, this model proved to be suitable for the testing of local antimicrobial strategies.

## 5. Limitations

This study has several limitations. Firstly, this study is valid for early-stage testing of antimicrobial biomaterials targeting infection prophylaxis. Therefore, implant and tissues were harvested on day 3 post infection. However, additional models are needed to evaluate the novel antimicrobial biomaterials for infection therapy at later time points. Secondly, cefazolin and gentamicin serum concentrations were not measured, which could have revealed insufficient levels and explained persistently infected animals. Only one type of implant was used with a uniform site over all included animals. The type of material, shape and size of the implant might have an impact on infection development and antibiotic efficacy. In future studies, other clinically relevant bacteria, especially Gram-negative, should also be investigated to enhance clinical relevance of this model. Lastly, as with every animal model, the pharmacokinetics and pharmacodynamics must be considered when extrapolating these results to humans.

## 6. Conclusions

This study presents a reproducible mouse infection model, which can be adapted for future research on novel strategies in the prevention and treatment of ODRI. Gentamicin-loaded PMMA discs with or without the additional systemic administration of cefazolin at a dose of 250 mg/kg proved highly effective in infection prevention and provide a suitable positive control to test other prevention strategies.

## Figures and Tables

**Figure 1 biomedicines-11-00040-f001:**
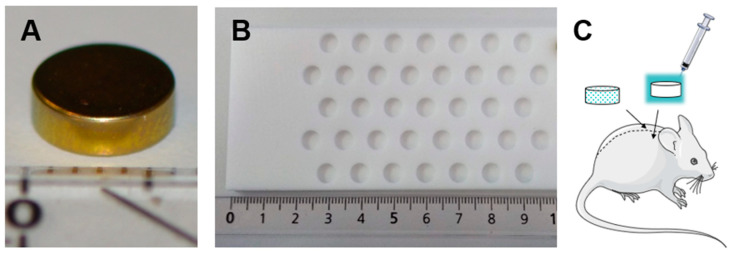
Implant design and implantation. (**A**) Titanium implant used in the study. (**B**) By using a custom-made Teflon mold, PMMA discs of the same shape and size as the titanium discs were obtained for the third phase. (**C**) Illustration of the two implants placed bilateral of the spine during the first and second phase. The implant on the left (dotted) represents the pre-inoculated implant. The box around the sterile implant demonstrates the inoculum pipetted (syringe) after implantation. In the third phase PMMA discs or titanium implants were inserted unilaterally, and the inoculum was pipetted.

**Figure 2 biomedicines-11-00040-f002:**
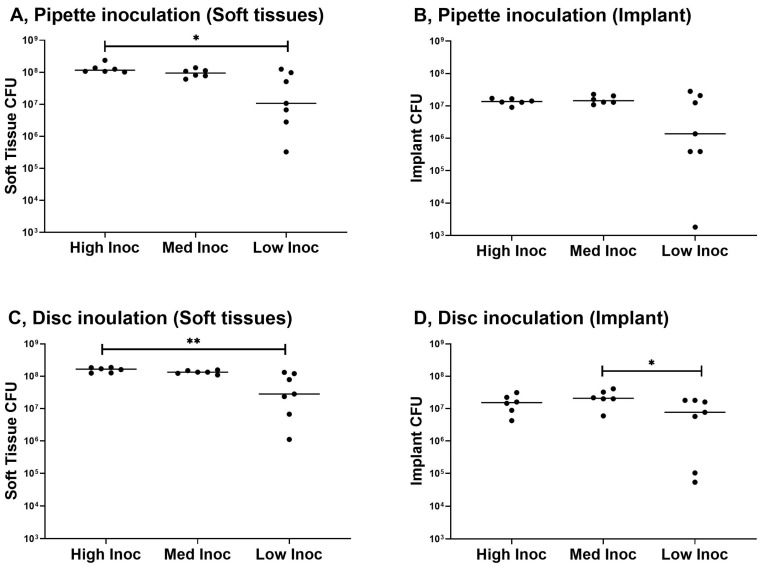
Results of dose-finding phase. Results are grouped according to the sample type and inoculation method for the high (n = 7), medium (n = 6) and low (n = 7) inoculum; (**A**) Pipette inoculation, soft tissue sample, * statistical significance between the high and low inoculum (*p* = 0.02); (**B**) Pipette inoculation, implant sample; (**C**) Disc inoculation, soft tissue sample, ** statistical significance between the high and low inoculum (*p* = 0.004); (**D**) Disc inoculation, implant sample, * statistical significance between the medium and low inoculum (*p* = 0.05). The detected CFU count is displayed on the *y*-axis; the three utilized bacterial inocula are represented on the *x*-axis.

**Figure 3 biomedicines-11-00040-f003:**
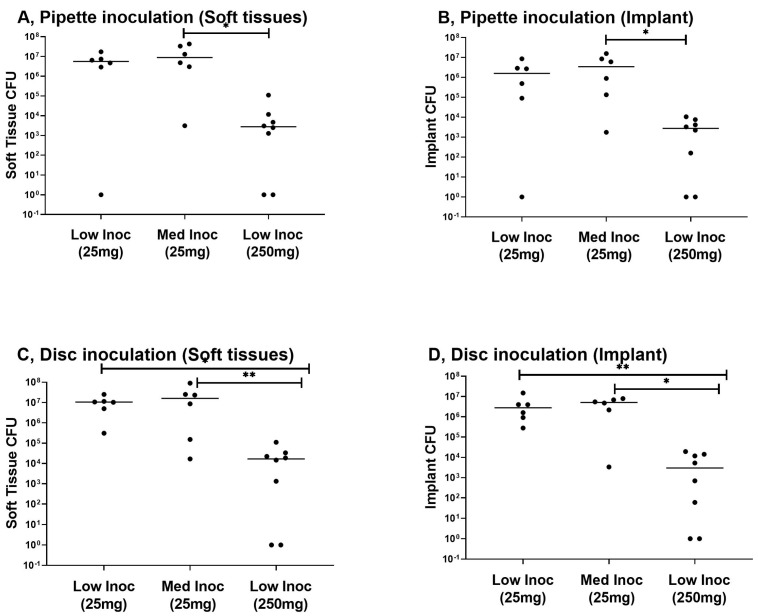
Results of systemic antibiotic prophylaxis phase. Results are grouped according to inoculation method and sample type for the low inoculum, treated with 25 mg (n = 6) or 250 mg (n = 8) of cefazolin and the medium inoculum, treated with 25 mg of cefazolin (n = 6). (**A**) Pipette inoculation, soft tissue sample, * statistical significance between the low inoculum treated with 250 mg of cefazolin and the medium inoculum treated with 25 mg of cefazolin (*p* = 0.01); (**B**) Pipette inoculation, implant sample, * statistical significance between the low inoculum treated with 250 mg of cefazolin and the medium inoculum treated with 25 mg of cefazolin (*p* = 0.03); (**C**) Disc inoculation, soft tissue sample, * statistical significance between the low inoculum treated with 250 mg of cefazolin and the low inoculum treated with 25 mg of cefazolin (*p* = 0.01), ** statistical significance between the low inoculum treated with 250 mg of cefazolin and the medium inoculum treated with 25 mg of cefazolin (*p* = 0.02); (**D**) Disc inoculation, implant sample, * statistical significance between the low inoculum treated with 250 mg of cefazolin and the medium inoculum treated with 25 mg of cefazolin (*p* = 0.01), ** statistical significance between the low inoculum treated with 250 mg of cefazolin and the low inoculum treated with 25 mg of cefazolin (*p* = 0.02). The CFU count is reflected on the *y*-axis; the *x*-axis displays inoculum and antibiotic concentration being used.

**Figure 4 biomedicines-11-00040-f004:**
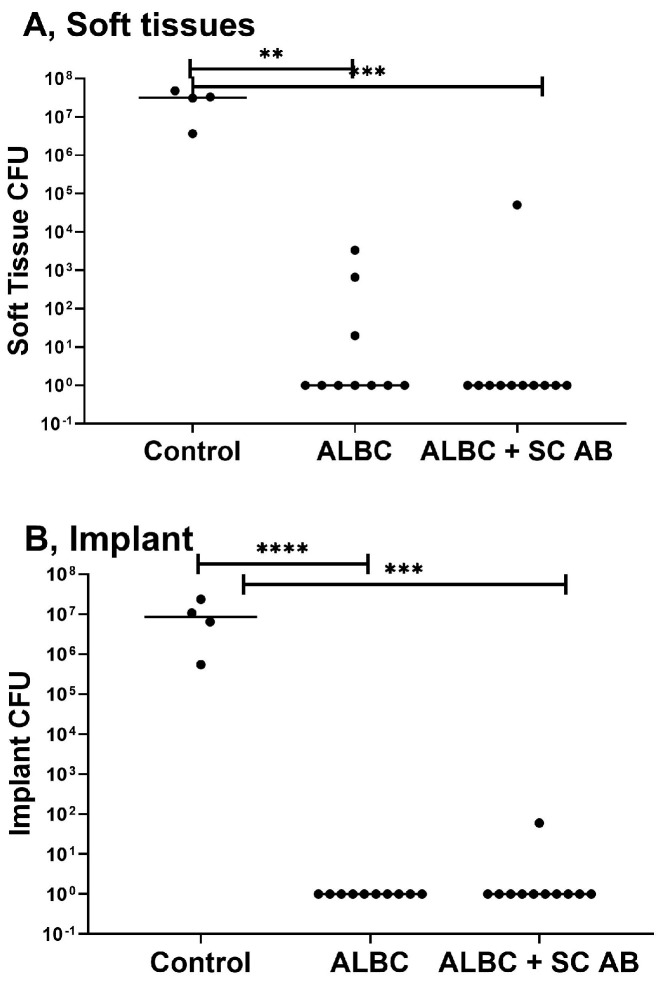
Results of local antibiotic prophylaxis phase. Results are grouped according to sample type for the control group (n = 4) and groups with ALBC with (n = 10) or without (n = 11) additional antibiotics. (**A**) Soft tissue sample, ** statistical significance between the control group and group with ALBC without additional antibiotics (*p* = 0.004), *** statistical significance between the control group and group with additional antibiotics (*p* = 0.0005); (**B**) Implant sample, *** statistical significance between the control group and group with additional antibiotics (*p* = 0.0002), **** statistical significance between the control group and group with ALBC without additional antibiotics (*p* < 0.0001). The CFU count is presented on the *y*-axis. The *x*-axis is reflecting the implant type, as well as additionally administered systemic antibiotic prophylaxis.

**Table 1 biomedicines-11-00040-t001:** Overview of study design.

Group	Prophylaxis	Inoculation Dose ^1^	Inoculation Method	Implant(s)/Animal	Group Size
**Dose-finding phase**					
1	None	High * Medium Low	Pre-adhered/ pipetted	2 titanium discs	7 6 7
**Systemic antibiotic prophylaxis phase**					
2 3	Cefazolin 25 mg/kg Cefazolin 250 mg/kg	Medium Low Low	Pre-adhered/ Pipetted Pre-adhered/ pipetted	2 titanium discs 2 titanium discs	6 6 8
**Local antibiotic prophylaxis phase**					
4 5 6	None Cefazolin 250 mg/kg None	Low Low Low	Pipetted Pipetted Pipetted	1 ALBC 1 ALBC 1 titanium disc	10 11 4
**Total**					65

^1,^* High, 1 × 10^6^ CFU/mL; Medium, 1 × 10^4^ CFU/mL; Low, 1 × 10^3^ CFU/mL, ALBC, Antibiotic Loaded Bone Cement.

## Data Availability

Not applicable.

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
