# Peer review of "Development and Characterization of a Subcutaneous Implant-Related Infection Model in Mice to Test Novel Antimicrobial Treatment Strategies"

_biomedicines, 2022, doi:10.3390/biomedicines11010040_

Round 1
Reviewer 1 Report
Dear Editor, Dear Authors,
I was invited to evaluate the manuscript « Development and characterization of a subcutaneous implant-related infection model in mice to test novel antimicrobial treatment strategies » by Wittmann et al.
In their study, the authors investigated a mouse model of subcutaneously implanted titanium discs, infected with S. aureus, as model of orthopaedic associated infection. For that, the authors developped different protocoles of inoculation before testing the effect of antibiotics administred through iv on the infected mice. The authors then evaluated antibiotic Loaded Bone Cement (ALBC) discs in term of infection prevention. Results obtained prove that ALBC discs with or without the additional antibiotic treatment are effective in preventing infection and allowed the authors to conclude their model scan be used to evaluate antimicrobial materials.
I found the study well designed and conducted. Results are clearly presented and convincing. I believe the model developped by the authors is important for the field.
I will have few comments.
- Although I totally understand that the authors used S aureus as model bacteria and it is already enoug to validate their model, would the authors have data (even preliminary ones) with gram-negative bacteria that could also be involved in orthopedic infection ?
- Although it may be hard, would the authors have try to develop a model in which the implants are inserted into bones like in the real situation in orthopedic surgery ? The fact that implants are inserted into bones may change the results, although bone implants will be harder to inoculate.
- Although the authors state dit in the limitations section of the manuscript, it would be good to have performed not only 3 days but also at least one longer incubation period. Does 3 days period is classical in this type of model ?
- Although I understand the point do focuse the bacterial analysis on the implants and soft tissues surrounding it, would the authors have data regarding systemic spreading of the bacteria (bacterial blood count) ?
regards
Author Response
Dear reviewer
thank you for your valuable input and comments to our study. The answer to your question are as following:
- Although I totally understand that the authors used S aureus as model bacteria and it is already enough to validate their model, would the authors have data (even preliminary ones) with gram-negative bacteria that could also be involved in orthopedic infection ?
We thank the reviewer for this clinically important and valid input. We agree that the inclusion of other bacteria, especially gram negative, would enhance the validity of our model. Unfortunately, at the moment we have not conducted further research investigating other bacterial strains (or several different ones) - but we plan to do so in the near future using this model.
We have added the following sentence to the manuscript (line 345 to 347): . "In future studies, other clinical relevant bacteria, especially gram negative, should also be investigated to enhance clinical relevance of this model." - Although it may be hard, would the authors have try to develop a model in which the implants are inserted into bones like in the real situation in orthopedic surgery ? The fact that implants are inserted into bones may change the results, although bone implants will be harder to inoculate.
This study was designed to describe and validate a low complexity model used for initial screening of novel antibacterial treatments:. We agree with the reviewer that in a second step a more complex model should be chosen where an implant directly onto the bone could be used to reflect better the clinical situation. In our opinion, the choice of model (and its complexity) does depend on the research question and we are convinced that the described model will be very useful for the early-stage testing of antimi-crobial biomaterials. We have addressed these points in the introduction line 56 to 68. - Although the authors state it in the limitations section of the manuscript, it would be good to have performed not only 3 days but also at least one longer incubation period. Does 3 days period is classical in this type of model ?
Thank you for pointing out this weak aspect of our manuscript. The study was designed to describe a model of infection prophylaxis. Infection prophylaxis is expected to have an immediate effect. To investigate the infection treatment a different model and timepoints would be needed. In our opinion, the original text was confusing regarding this point. We have reworded our statement within 5. Limitations (Iine 336 to 341): "Firstly, this study is valid for early-stage testing of antimicrobial biomaterials targeting infection prophylaxis. Therefore, implant and tissues were harvested on day 3 post infection. However, additional models are needed to evaluate novel antimicrobial biomaterials for infection therapy at later time points. - Although I understand the point do focus the bacterial analysis on the implants and soft tissues surrounding it, would the authors have data regarding systemic spreading of the bacteria (bacterial blood count)
All animals were monitored postoperatively twice a day by a veterinarian. No clinical signs of systemic infection were observed. Therefore, we think systemic spread of bacteria is very unlikely, but can`t be excluded. We added the following sentence (line 197): " ...and showed no clinical signs of systemic infection."
We hope to have addressed all raised points appropriately and again, we appreciate the reviewers time and valuable input on the study.
Best regards
Stephan Zeiter
Reviewer 2 Report
The ms: Development and Characterization of a Subcutaneous Implant- Related Infection Model in Mice to Test Novel Antimicrobial Treatment Strategies, presents the design and characterization of a subcutaneous implant-associated infection model in mice to fill that gap in the early-stage testing of antimicrobial biomaterials. The model is well explained and correlated with the results and conclusions, but could be suggested that the topic goes further beyond conventional, because in this way could be obtained a novel contribution in this area. If the authors deep in the mechanism why microorganisims are adapted to get resistance could be quite original and novel.
Author Response
Dear reviewer
thank you for your valuable input and comments to our study. We agree that it is very important, novel and interesting to look into the mechanism why microorganism can fight current treatment strategies. Actually, this study presents the design and characterization of a model in mice for early-stage testing of antimicrobial biomaterials. These biomaterials may target the mentioned bacterial mechanism of resistance. However, to study these bacterial mechanism was not the goal of the study.
Again, we appreciate the reviewers time and valuable input on the study.
Best regards
Stephan Zeiter